# Dissolved Gas Analysis Equipment for Online Monitoring of Transformer Oil: A Review

**DOI:** 10.3390/s19194057

**Published:** 2019-09-20

**Authors:** Sergio Bustamante, Mario Manana, Alberto Arroyo, Pablo Castro, Alberto Laso, Raquel Martinez

**Affiliations:** School of Industrial Engineering, University of Cantabria, Av. Los Castros s/n, 39005 Cantabria, Spain; bustamantes@unican.es (S.B.); arroyoa@unican.es (A.A.); castropb@unican.es (P.C.); lasoal@unican.es (A.L.); raquel.martinez@unican.es (R.M.)

**Keywords:** dissolved gas analysis, power transformer, transformer maintenance, transformer oil

## Abstract

Power transformers are the most important assets of electric power substations. The reliability in the operation of electric power transmission and distribution is due to the correct operation and maintenance of power transformers. The parameters that are most used to assess the health status of power transformers are dissolved gas analysis (DGA), oil quality analysis (OQA) and content of furfuraldehydes (FFA) in oil. The parameter that currently allows for simple online monitoring in an energized transformer is the DGA. Although most of the DGA continues to be done in the laboratory, the trend is online DGA monitoring, since it allows for detection or diagnosis of the faults throughout the life of the power transformers. This study presents a review of the main DGA monitors, single- or multi-gas, their most important specifications, accuracy, repeatability and measurement range, the types of installation, valve or closed loop, and number of analogue inputs and outputs. This review shows the differences between the main existing DGA monitors and aims to help in the selection of the most suitable DGA monitoring approach according to the needs of each case.

## 1. Introduction

The most important and expensive asset in power transmission and distribution networks is the power transformer, so there must have a well-defined maintenance strategy from commissioning to withdrawal to ensure an appropriate level of reliability throughout the operational life of the transformer.

According to [1,2,3], different asset maintenance strategies coexist depending on the condition and available information of the components and subcomponents of the substation equipment, as seen in Figure 1. The current trend in maintenance strategies for substation assets is to maintain a predictive maintenance approach based on prognostics, understanding that the prognostic tools do not strictly assess the remaining operating time but can be used to assess the future degradation of an asset [1].

In predictive maintenance strategies, the lifetime management of high-voltage substation equipment is pursued [1], with the aim to manage the risks of the assets of a substation. To perform lifetime management of the transformers, it is necessary to have all the possible data throughout the lifetime of the transformer, which can be obtained through condition-based maintenance and through online continuous monitoring technologies [1,4,5,6,7,8,9,10,11,12].

According to [13], the risk is defined as the “effect of the uncertainty in the objectives,” so risk management of the assets has the purpose to evaluate, create and protect the correct operation of assets throughout the lifetime. The risk of an asset is represented by the risk index. The risk index is a function of the probability of failure and the consequences of failure [1,5,6,14]. The probability of failure refers to the health index of an asset, while the failure consequence evaluates and defines the consequences of a failure event.

## 2. Health Index of the Power Transformer

The GB Distribution Network Operators (DNO) created a Network Asset Indices Methodology [14] that aims to create a common framework of definitions, principles and methodologies for calculating the health index (HI). This report is adopted by all GB DNO.

From [14], it was observed that more parameters are used in calculating the transformer HI than for the HI calculation of any other asset; for this reason, it was necessary to establish an order of importance for the parameters that are used. All condition criteria that can be used to calculate the transformer HI [15] are shown in Table 1. Figure 2 aims to show the location of the parameters, which are measured for the HI calculation of the transformer.

In a previous study [15,16,17,18,19,20,21,22,23,24,25,26,27,28,29,30,31], the parameters that were used in their equations or algorithms have different weights. These weight differences, equations and algorithms between studies are discussed in [32], and this review assumes that the subjective judgement of the weighting factor leads to different values of the final HI. Power transformers, mainly above 30–40 MVA, are custom designed machines [33], so for the HI calculation, it is necessary to take into account the differences of the transformers in the limits and weights that are applied in the equations and algorithms.

Thus, the best way to calculate the transformer HI is to always know as many parameters as possible (Table 1); however, although all the parameters are important, they are often partially unknown, difficult to collect or difficult to automatically incorporate into computer programs. For example, the thermal images of the transformer cannot be included easily in computer programs for the calculation of the transformer HI. Because an expert is necessary to interpret the thermal images which makes the calculation of the transformer HI slower. Therefore, the most used and most important parameters for calculating the transformer HI are dissolved gas analysis (DGA), oil quality analysis (OQA) and the content of furfuraldehydes (FFA) in the oil.

DGA: The DGA method measures the gas concentrations in oil that are formed by the insulation decomposition processes, which occur when the transformer has faults. Depending on the type of fault, different types of decomposition processes can occur. When electrical and thermal defects occur in the transformer oil, they degrade generating combustible gases, such as hydrogen ($H_{2}$), ethylene (C${}_{2}$H${}_{4}$), acetylene (C${}_{2}$H${}_{2}$), methane (CH${}_{4}$) and ethane (C${}_{2}$H${}_{6}$). When decomposition occurs in cellulosic insulation, the generated gases are carbon monoxide (CO) and carbon dioxide (CO${}_{2}$), and these gases indicate a thermal fault. Depending on the gas concentration that is measured, the type of fault can be identified by using the interpretation method that was collected in [34,35].OQA: The OQA consists of a combination of electrical, physical and chemical tests. The list of all the tests that can be performed on the transformer oil is shown in IEEE Std C57.106 [4]. The most important and common are the dielectric breakdown voltage (BDV), the water content, the power factor, the interfacial tension (IFT), acidity and colour. The results of these tests are used to prevent incipient failures and to evaluate the preventive maintenance processes, such as the replacement or recovery of transformer oil [36]. Even the use of these tests are different and have different weights when calculating the HI, depending on the study [15,16,17,18,19,20,21,22,23,24,25,26,27,28,29,30,31].FFA: The content of FFA in the transformer oil indicates the decomposition processes of the cellulosic material that constitute the transformer solid insulation [37]. The furanic components remain adsorbed by the paper, while a small part is dissolved in the oil. Its presence in the oil is used to diagnose the equipment in service as a complementary information to the DGA. Although the content of FFA in the transformer oil is a very important parameter in the calculation of HI, there are no recommendations for the interpretation of the results in the standards, as indicated by [24,28,38,39], so in each of the studies [15,16,17,19,20,22,23,26,30] a different limit value is taken in the HI calculation.

## 3. Dissolved Gas-In-Oil Analysis

Given the most important parameters in calculating the transformer HI defined above, the parameter that currently allows simple online monitoring of an energized transformer is the DGA [5,6,7,8,40].

DGA can be applied to various types of insulation oil used to fill the transformer [34,35,41,42,43,44]. Aging, thermal properties of each oil as well as the methods for the faults identification in the transformer oil are different depending on the oil used to fill the transformer. This review focuses on transformers filled with mineral oil and DGA equipment for online monitoring that measure the gas concentration in this type of oil.

Accurate online DGA monitoring makes it possible to detect or diagnose, almost instantaneously, any incipient failure that occurs in the liquid or solid insulation of the transformer, avoiding a major failure.

The use of gas concentrations limits and the methods for faults identification defined in the standards [34,35] allow for the detection and identification of early failures.

The gases that best detect an incipient fault are hydrogen and carbon monoxide. According to the gas generation based on temperature [45] and the key gas method (KGM) [34], as seen in Figure 3 and Figure 4.

Hydrogen is present, in higher or lesser amounts, in all the electrical faults and thermal faults that occur in the oil. Figure 3 shows the approximate generation of combustible gases based on temperature. The band on the left shows the approximate proportions of gases that are generated under partial discharge conditions; as shown, the amount of hydrogen is much greater than the rest of the gases. From the gas generation chart, it can be observed how hydrogen is present in all temperatures, from 150 to >800 ${}^{\circ }C$, while the rest of the gases, such as ethylene or acetylene, need high temperatures, such as from 350 to 500 ${}^{\circ }C$, respectively, to be generated; or in very high temperatures, gases such as methane and ethane are not generated.

Carbon monoxide is produced when the solid insulation is decomposed due to thermal faults according to KGM (Figure 4).

Therefore, a monitoring process aimed at detecting incipient faults in the transformer should at least measure the concentrations of hydrogen and carbon monoxide. As seen below, all DGA monitors measure oil moisture, so the measurement of the carbon monoxide concentration is not essential, since the degradation of the solid insulation is indicated by the increase of oil moisture. The measurement of these three concentrations (hydrogen, carbon monoxide and moisture) would allow a basic initial diagnosis, by differentiating between thermal and electrical faults or leaks into oil, according to Table 2.

There are many methods for the faults identification in the transformer insulation [34,35,47], in Table 3 the main characteristics of the most used methods are summarized.

Several studies [48,49,50,51,52,53,54,55,56] indicate that the best method for the faults identification in the transformer insulation is the Duval triangle method (DTM).The Duval pentagon method (DPM) was created to improve DTM results. Several recent studies [48,49] indicate that the DPM success rate improves those of the DTM, since the DPM allows to identify the normal aging of the transformer insulation. Although the DPM is shown as the best method of identifying faults, it should be noted that it is a fairly new method that should be studied further to improve its validity. So the DTM is the method considered as the best and most established to diagnose faults in the transformer insulation.

The DTM, as seen in Figure 5, uses the concentration ratio of three combustible gases (acetylene, ethylene and methane) to identify the fault. The concentration ratios of the three gases are calculated as
(1)$$\begin{array}{cc} \%C_{2}H_{2}= & \frac{100\cdot x}{x+y+z}\\ \%C_{2}H_{4}= & \frac{100\cdot y}{x+y+z}\\ \%{\mathrm{CH}}_{4}= & \frac{100\cdot z}{x+y+z} \end{array}$$
where *x*, *y* and *z* are gas concentrations in ppm of acetylene, ethylene and methane, respectively.

## 4. DGA Monitors

Online gas-in-oil monitors measure the gas concentrations in the transformer oil and detect or identify transformer faults.

CIGRE made a report on gas monitors for oil-filled electrical equipment [57] in which a list of the main monitors available in 2007 and the most important specifications were shown.

Table 4 shows the list of online gas-in-oil monitors and the main specifications that were analysed in this study. These monitors measure up to nine gas concentrations (hydrogen, ethylene, acetylene, methane, ethane, carbon monoxide, carbon dioxide, oxygen, nitrogen, propane or propene) from a single gas concentration (hydrogen). The amount of gas concentrations that this equipment can measure determines their functionality.

The equipment that measures from one to two gas concentrations is a fault detection monitor, known as single-gas DGA. Normally, these monitors measure hydrogen and carbon monoxide concentrations, which are the gases that indicate thermal and electrical faults in the transformer oil and thermal faults in the solid insulation [34]. There are equipment that also measure acetylene, ethylene or carbon dioxide, in addition to measuring hydrogen and carbon monoxide; although these monitors measure more gas concentrations, they do not identify faults. The fault detection monitor is used to monitor abnormal gassing in the transformer oil.

The equipment that measures nine gas concentrations is the fault diagnosis monitor, known as a multi-gas DGA. Using the values of gas concentrations and the methods defined in the standards [34,35], these monitors identify transformer faults. The fault diagnosis monitor is used to monitor the abnormal gassing in the transformer oil and diagnose faults.

As shown in Table 4, each manufacturer of DGA monitors uses a different technology for the detection and diagnosis of dissolved gases in the online monitoring transformer oil. The main manufacturers use predominantly gas chromatography (GC), photoacoustic spectroscopy (PAS), solid-state (IC), thermal conductivity detector (TCD), non-dispersive infrared (NDIR), infrared (IR), near infrared (NIR), Fourier transform infrared (FTIR), fuel cell (FC), micro-electronic sensor or electrochemical cell [58,59,60,61,62,63,64,65,66,67,68,69,70,71,72,73,74,75,76,77,78,79,80,81,82,83,84,85]. Several studies [86,87,88,89,90,91] present a review of these gas sensor technologies for the detection of different gases.Table 5 shows the advantages and disadvantages collected in [86,87,88,89,90,91] of the gas detection methods of the DGA online monitoring equipment presented in Table 4.

According to the report on gas-in-oil monitors published by CIGRE [57], most manufacturers use the gas extraction method based on the headspace principle [92,93,94]. In this study, all monitors use the gas extraction method based on the headspace principle by direct contact between the oil and a small gas phase above or through a membrane separating the two phases (membrane of semipermeable PTFE or another polymer), as shown in Table 4. The extraction of gases is performed under different conditions of pressure (atmospheric or under partial vacuum) and temperature (at oil temperature, at ambient temperature, or at a fixed temperature), with or without pumping the oil continuously on the monitor. Once the gas extraction is performed, each gas measurement technology [86,87,88,89,90,91,92] measures the gas concentrations present in the oil sample.

Monitors that use GC measurement technology need carrier and calibration gases. Helium is used as a carrier gas to transport the sample gases that are extracted from the transformer oil. Gas chromatography monitors perform automatic calibration using on-board National Institute of Standards and Technology (NIST) traceable calibration gas. Table 4 shows the elapsed time between automatic calibrations, depending on the manufacturer.

Monitors that use the fuel cell as a measurement technology need to replace the sensor every several years and perform sensor calibration every time the sensor is replaced.

### 4.1. Installation of The Monitor

The installation of the online gas-in-oil monitors is different depending on the type of sensor and transformer [95]. According to the manufacturers [58,59,60,61,62,63,64,65,66,67,68,69,70,71,72,73,74,75,76,77,78,79,80,81,82,83,84,85], the installation of the monitors is performed using one or two different valves of the transformer. Table 4 shows the type of installation used by each sensor that is analysed in this study, and Figure 6a shows the possible valve locations on a power transformer.

The installation of the monitor using a single valve is performed by mounting the monitor directly on the valve using a thread or a flange (Figure 6b). In this type of installation, the best place to install the monitor is the valve located in the straight section in the cooling loop outlet pipe (Figure 6a), because there is oil flow, which makes the sample representative. An alternative installation, if the previous one is not available, consists of using a valve far enough from the bottom to ensure adequate oil flow, like the fill valve (Figure 6a). The drain valve is the worst location to install the monitor because it is located in the bottom of the transformer, where there is no oil flow and there is a risk of oil sludge.

The installation of the monitors using two valves consists of creating a closed loop in which the oil passes through the monitor (Figure 6b). One of the valves is the supply valve and the other is the return valve. As in the case of single valve installation, the supply valve must take oil where there is oil flow, and the valve that is located in the straight section in the radiator outlet pipe, the fill valve or a valve that is far enough from the bottom are good choices. The return valve can be the drain valve or an auxiliary valve whenever it is below the supply valve.

Two types of installation (Figure 6b) have several differences, apart from the number of valves they use. In the installation of the monitor in a valve, the oil sample is taken from and returned to the same valve, and the sensor or sensors are in this area where there is only circulation of the oil inside the tank. Other monitors with the installation in a valve generate oil circulation in the valve area through the heating of a pipe inside the valve to obtain a better oil flow. Although the best and most representative oil flow is achieved by using an oil loop installation, in addition to the internal flow in the tank, new flow is added due to this loop, thus making the oil sample in the monitor as representative as possible.

### 4.2. Fault Detection Monitor (Single-Gas DGA)

Fault detection monitors send out warnings when the gas concentration exceeds a set limit, which is normally the limits of each gas used are those indicated by the IEEE [34] and IEC [35] standards, or are calculated using the methods or recommendations proposed in [41,96,97,98,99].

Table 6 shows the gas concentrations that were measured by the fault detection monitors; it also shows the monitors that measure the oil moisture.

Table 7, Table 8 and Table 9 show the range, accuracy and repeatability of the measurements of hydrogen, carbon monoxide and moisture concentrations by the fault detection monitors, respectively, according to the manufacturers [58,61,63,64,65,70,72,74,75,78,79,81,82].

The measurement range indicates the lower detection limit (LDL) and the upper detection limit (UDL) in parts per million (ppm) that the monitor measures. The LDL is the minimum gas concentration that the monitor is able to measure; cases in which the manufacturers indicate a lower measurement range equal to 0 ppm does not indicate that the LDL is 0 ppm, but measurements below the LDL will be considered to be 0 ppm. The LDL is the most important parameter in the measurement range, because it allows for the detection of changes at very low concentrations. The accuracy and repeatability are shown in ppm and in percentage (%); the highest value is always taken.

The lower the ppm and the percentage of accuracy and repeatability and the lower the LDL ppm from Table 7, Table 8 and Table 9, improve fault detection monitoring [91].

In observing the moisture measurement values (Table 9), all fault detection monitors had similar specifications, with differences approximately 1 ppm or 1% in accuracy and 1–2% in the measurement range.

### 4.3. Fault Diagnosis Monitor (Multi-Gas DGA)

Fault diagnosis monitors, in addition to sending out warnings when the gas concentration exceeds a set limit, also diagnose the fault using the standards of IEEE [34] and IEC [35].

Table 10 shows the gases measured by the fault diagnosis monitors; it also shows the monitors that measure the oil moisture.

Table 11 shows the range, accuracy and repeatability of the gas concentration measurements by the fault diagnosis monitors according to the manufactures [59,60,62,66,67,68,69,71,73,76,77,80,83,84,85].

As in the case of the fault detection monitors, accuracy and repeatability are different depending on the monitor and the manufacturer, and the accuracy and repeatability in the moisture measurement are very similar among monitors. Accuracy and repeatability are the main characteristics to be observed in the diagnosis and online monitoring of transformer oil to obtain reliable results.

### 4.4. Analogue Inputs and Outputs of DGA Monitors

As mentioned above, online monitoring of power transformers allows having an asset maintenance strategy oriented to predictive maintenance. Most gas-in-oil monitors have analogue inputs and outputs whereby different external sensors can be connected to monitor other asset conditions. These external sensors measure, among others, the ambient temperature, oil temperature, moisture-in-oil and loading conditions.

All the monitors that have analogue outputs use DC current ($I_{DC}$) in the 4–20 mA range for their analogue outputs. In the case of analogue inputs, most monitors also use the DC current in the 4–20 mA range, but there are several monitors that also use the AC current ($I_{AC}$), AC voltage ($V_{AC}$) or DC voltage ($V_{DC}$), in the ranges of 4–20 mA + 20%, 0–80 V + 20% and 0–10 V + 20%, respectively.

Table 12 shows the analogue inputs and outputs of the fault detection monitors and the fault diagnosis monitors.

## 5. Conclusions

The dissolved gas analysis of the transformer oil is one of the most important parameters when evaluating the health status of a power transformer. Online DGA monitoring aims to support predictive maintenance of the power transformers by detecting incipient faults in the liquid or solid insulations.

A review of DGA equipment for online monitoring of power transformers was performed in this study. The main monitors that were available on the date of writing of this study (2018) were presented. A classification of the types of existing DGA monitors was defined in this study of fault detection monitors and fault diagnosis monitors. Fault detection monitors measure the concentrations of hydrogen and carbon monoxide that are present in the transformer oil, so they only detect the fault, whereas fault diagnosis monitors measure the concentrations of five to nine gases, so they are able to diagnose the type of fault. Apart from the amount of gases they measure, DGA monitors have differences in accuracy, repeatability and measurement range. DGA monitors have two types of installation, in a valve or in a closed loop. Finally, each monitor has a different number of analogue inputs and outputs to connect to other sensors and to obtain better monitoring of the asset.

Another point to consider in the selection of a fault diagnosis monitor is the method to be used for fault identification. Depending on the number of gases required for faults identification of the method to be used, the selection of the monitor is limited. The comparison between the main methods for the faults identification, type of faults identifiable and gas concentrations required by each method was shown in this review.

This review aims to help the selection of DGA monitors, fault detection or diagnosis monitors, the required specifications depending on the importance of the transformer, the number of valves needed by the monitor and the number of valves in the transformer and the need to connect other sensors to the monitor through analogue inputs, or to connect the analogue outputs of the monitor to other equipment.

## Figures and Tables

**Figure 1 sensors-19-04057-f001:**
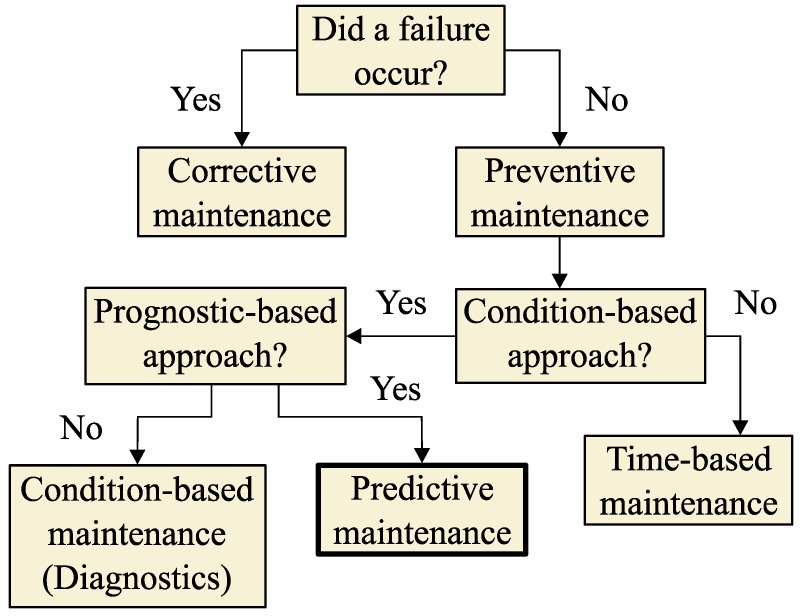
Interrelation of maintenance strategies [1].

**Figure 2 sensors-19-04057-f002:**
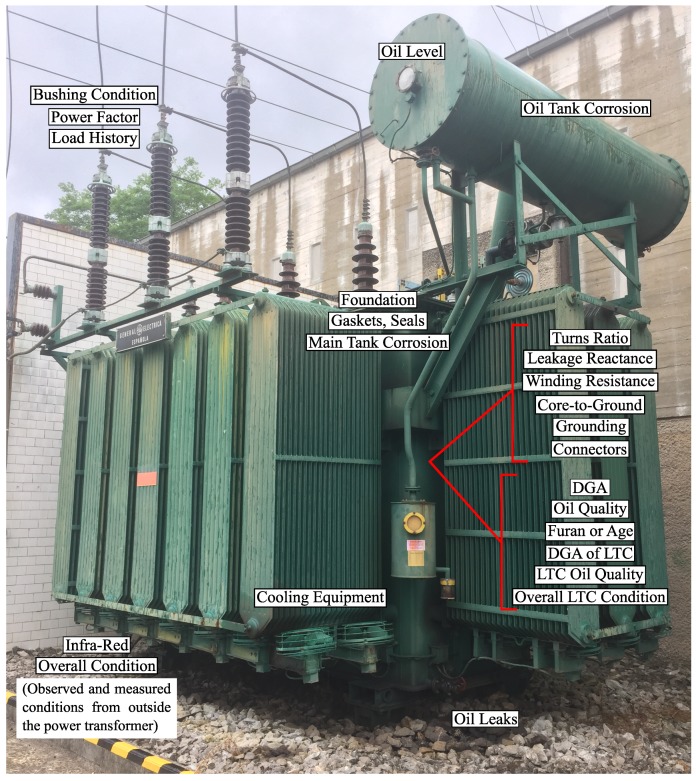
Location of the parameters measured for the HI calculation of the transformer.

**Figure 3 sensors-19-04057-f003:**
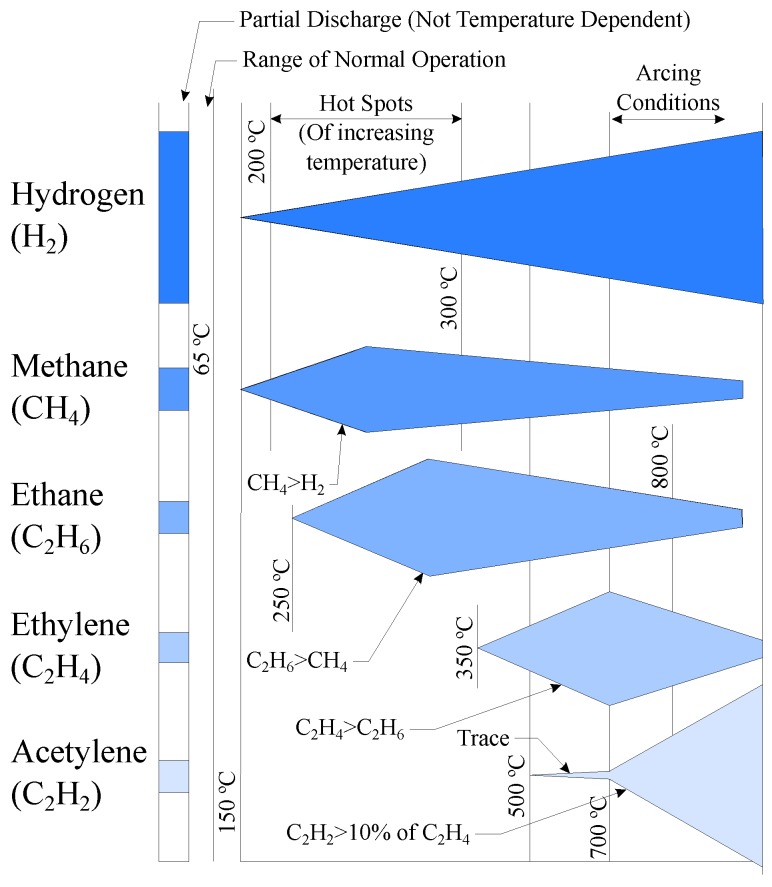
Gas generation based on temperature (Not to scale) [45].

**Figure 4 sensors-19-04057-f004:**
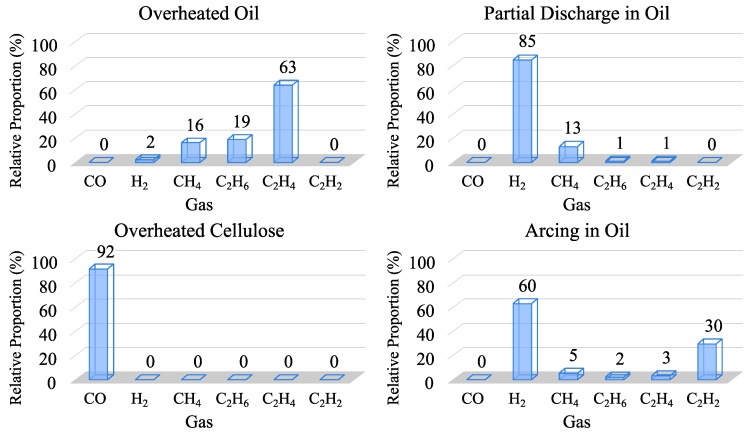
Fault type by the key gas method [34].

**Figure 5 sensors-19-04057-f005:**
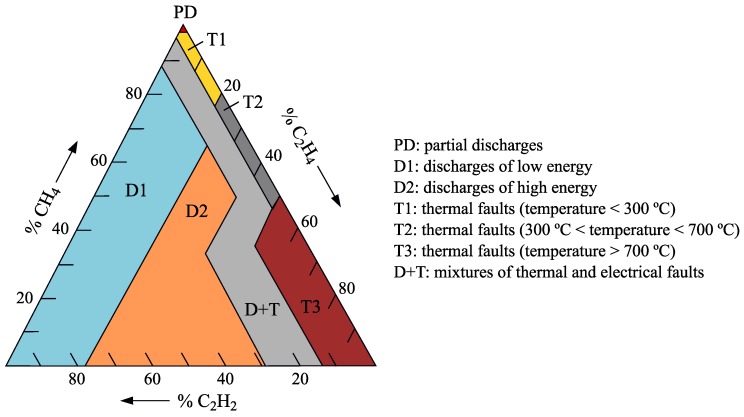
Duval Triangle and list of faults detectable by DGA [37].

**Figure 6 sensors-19-04057-f006:**
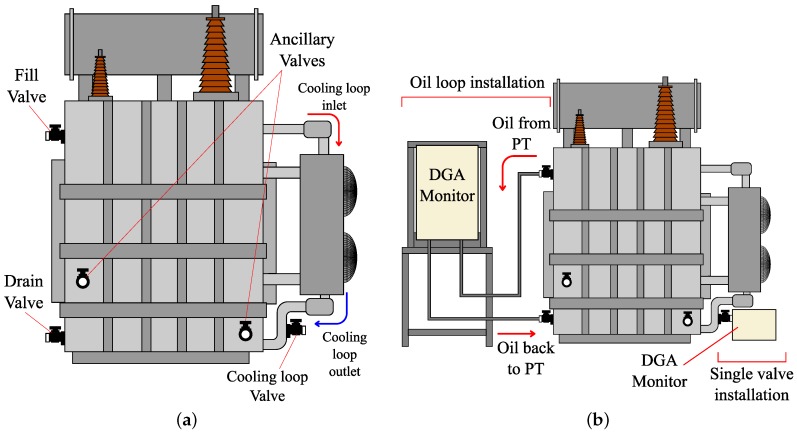
(**a**) Possible valve locations in a power transformer; and (**b**) installation schematic (Not to scale).

**Table 1 sensors-19-04057-t001:** Parameters of the transformer health index [15].

Item	Condition Criteria	Item	Condition Criteria
1	DGA	13	Main Tank Corrosion
2	Load History	14	Cooling Equipment
3	Power Factor	15	Oil Tank Corrosion
4	Infra-Red	16	Foundation
5	Oil Quality	17	Grounding
6	Overall Condition	18	Gaskets, Seals
7	Furan or Age	19	Connectors
8	Turns Ratio	20	Oil Leaks
9	Leakage Reactance	21	Oil Level
10	Winding Resistance	22	DGA of LTC
11	Core-to-Ground	23	LTC Oil Quality
12	Bushing Condition	24	Overall LTC Condition

**Table 2 sensors-19-04057-t002:** Initial fault identification chart [46].

Fault	Gas Generated		
	CO	H${}_{2}$	H${}_{2}$O
Cellulose aging	x		x
Mineral oil decomposition		x	
Leaks into oil			x
Thermal decomposition of cellulose	x	x	
Overheated transformer core	x	x	
Thermal faults in oil (150 to 300 ${}^{\circ }$C)		x	
Thermal faults in oil 300 to 700 ${}^{\circ }$C)		x	
Thermal faults in oil (> 700 ${}^{\circ }$C)		x	
Partial discharge		x	
Arcing		x	

**Table 3 sensors-19-04057-t003:** Comparison between fault identification methods [34,35,47].

Method	Description	Fault Identification and Normal Aging	Gas Used
Key Gas Method (KGM)	Uses individual gas concentrations to identify the fault	PD, arcing, overheated oil, overheated cellulose	CO, H${}_{2}$, C${}_{2}$H${}_{2}$, C${}_{2}$H${}_{4}$
Doernenburg Ratio Method (DRM)	Uses four gas concentration ratios: $\frac{{\mathrm{CH}}_{4}}{H_{2}},\frac{C_{2}H_{2}}{C_{2}H_{4}},\frac{C_{2}H_{2}}{{\mathrm{CH}}_{4}},\frac{C_{2}H_{6}}{C_{2}H_{2}}$	Thermal decomposition, PD, arcing	H${}_{2}$, C${}_{2}$H${}_{2}$, CH${}_{4},$ C${}_{2}$H${}_{6}$, C${}_{2}$H${}_{4}$
Rogers Ratio Method (RRM)	Uses three gas concentration ratios: $\frac{{\mathrm{CH}}_{4}}{H_{2}},\frac{C_{2}H_{2}}{C_{2}H_{4}},\frac{C_{2}H_{4}}{C_{2}H_{6}}$	Normal aging, PD, arcing, low temperature fault, thermal fault <700 ${}^{\circ }$C, thermal fault >700 ${}^{\circ }$C	H${}_{2}$, C${}_{2}$H${}_{2}$, CH${}_{4},$ C${}_{2}$H${}_{6}$, C${}_{2}$H${}_{4}$
IEC Ratio Method (IRM)	Uses three gas concentration ratios: $\frac{{\mathrm{CH}}_{4}}{H_{2}},\frac{C_{2}H_{2}}{C_{2}H_{4}},\frac{C_{2}H_{4}}{C_{2}H_{6}}$	PD, low energy discharge, high energy discharge, thermal faults <300 ${}^{\circ }$C, between 300 and 700 ${}^{\circ }$C, and >700 ${}^{\circ }$C	H${}_{2}$, C${}_{2}$H${}_{2}$, CH${}_{4},$ C${}_{2}$H${}_{6}$, C${}_{2}$H${}_{4}$
Duval Triangle Method (DTM)	Uses three gases corresponding to the increasing energy content or temperature of the faults	PD, low energy discharge, high energy discharge, thermal faults <300 ${}^{\circ }$C, between 300 and 700 ${}^{\circ }$C, and >700 ${}^{\circ }$C	C${}_{2}$H${}_{2}$, CH${}_{4},$ C${}_{2}$H${}_{4}$
Duval Pentagon Method (DPM)	Uses five gases corresponding to the increasing energy content or temperature of the faults	Normal aging, PD, low energy discharge, high energy discharge, thermal faults <300 ${}^{\circ }$C, between 300 and 700 ${}^{\circ }$C, and >700 ${}^{\circ }$C	H${}_{2}$, C${}_{2}$H${}_{2}$, CH${}_{4},$ C${}_{2}$H${}_{6}$, C${}_{2}$H${}_{4}$

**Table 4 sensors-19-04057-t004:** List of manufacturers and gas-in-oil monitors.

Manufacturer	Equipment	Measurement Technology	Gas Extraction	Consumables	Automatic Calibration	Installation
Morgan Schaffer	Calisto 2 [58]	TCD	PTFE			2 V
	Calisto 5 [59]	GC	PTFE	Calibration and carrier gases	Every 24 hours	2 V
	Calisto 9 [60]	GC	PTFE	Calibration and carrier gases	Every 24 hours	2 V
LumaSense	SmartDGA Guard [61]	NDIR + FC	Membrane	H${}_{2}$ sensor	After sensor replacement	1 V or 2 V
	SmartDGA Guide [62]	NDIR + FC	Membrane	H${}_{2}$ and O${}_{2}$ sensor	After sensor replacement	1 V or 2 V
GE	Minitrans [63]	PAS	Headspace			2 V
	Hydran 201T*i* [64]	FC	Membrane			1 V
	Hydran M2-X [65]	FC	Membrane	H${}_{2}$ sensor		1 V
	Taptrans [66]	PAS	Headspace			2 V
	Transfix [67]	PAS	Headspace			2 V
	Multitrans [68]	PAS	Headspace			2 V
	DGA 900 [69]	PAS	Headspace			2 V
Vaisala	MHT 410 [70]	IC	Headspace			1 V
	OPT 100 [71]	IR	Headspace			2 V
ABB	CoreSense [72]	IC	Headspace			1 V
	CoreSense M10 [73]	IC + FTIR	Headspace			1 V
Qualitrol	DGA 150 [74]	IC	Headspace			1 V
	TM1 [75]	IC	Headspace			1 V
	TM3 [76]	GC	Membrane	Calibration and carrier gases	Every 3 days	2 V
	TM8 [77]	GC	Membrane	Calibration and carrier gases	Every 3 days	2 V
MTE	Hydrocal 1003 [78]	Micro-electronic sensor + Electrochemical cell	Membrane			1 V
	Hydrocal 1004 GenX [79]	Micro-electronic sensor + NIR	Membrane			1 V
	Hydrocal 1005 [78]	Micro-electronic sensor + NIR	Membrane			1 V
	Hydrocal 1008 [80]	Micro-electronic sensor + NIR	Membrane			1 V
	Hydrocal 1009 [80]	Micro-electronic sensor + NIR	Membrane			1 V
SIEMENS	SITRAM H2Guard [81]	IC	Headspace			1 V
	SITRAM Multisense 5 [82]	Micro-electronic sensor + NIR	Headspace			1 V
	SITRAM Multisense 9 [83]	Micro-electronic sensor + NIR	Headspace			1 V
CAMLIN	TOTUS G5 [84]	IR	–			2 V
	TOTUS G9 [85]	PAS	–			2 V

TCD = Thermal conductivity detector; GC = Gas chromatography; NDIR = Non dispersive infrared; FC = Fuel cell; PAS = Photoacoustic Spectroscopy; IC = Solid-state sensor; NIR = Near infrared; IR = Infrared; FTIR = Fourier-transform infrared; 1 V = Single valve installation; 2 V = Close loop installation; –: Not indicated by the manufacturer; Blank spaces indicate that monitors do not need consumables or calibration owing to the use or replacement of consumables.

**Table 5 sensors-19-04057-t005:** Advantages and disadvantages of gas detection technologies [86,87,88,89,90,91].

Technology	Advantages	Disadvantages
GC	Wide range of fault gases Highest accuracy and repeatability	Long time required to complete a test Expensive Frequent calibrations needed Auxiliary (carrier) gas needed Maintenance cost
PAS	Wide range of fault gases Can detect/measure very low (ppm and ppb) gas concentrations Low maintenance	Results are sensitive to the wave number range of the optical filters and their absorption characteristics Accuracy influenced by temperature, pressure, and vibration Limited ability to measure high gas concentrations Interfering gases can effect accuracy
IC	Operate under extreme temperatures, vibration, or in corrosive atmospheres	Limited ability to detect very low gas concentrations
TCD	Fast response Stable Wide measuring range Simple construction Robust	Sensitive to interfering gases Reaction due to heating wire Heating element reacts with gas
NDIR	Simultaneous multi-gas measurement No required calibrations Low maintenance Fast gas measurement time	Limited ability to detect very low gas concentrations Interfering gases can effect accuracy
IR	Uses only physical technique Can be used in inert atmospheres	Not all gases have IR absorption Sequential monitoring is slower on multi point analyzers and also more user expertise required
NIR	Simultaneous multi-gas measurement Non-frequent calibrations Low maintenance	Limited ability to measure high gas concentrations Interfering gases can effect accuracy
FTIR	Simultaneous multi-gas measurement	Accuracy influenced by moisture
FC	Small size	Periodic replacement Single gas measurement
Micro-electronic sensor	Small size	Single gas measurement
Electrochemical cell	Small size Working at high temperature is possible	Frequent calibrations needed Short/limited life time Single gas measurement Cross sensitivity to other gases

**Table 6 sensors-19-04057-t006:** Parameters measured by fault detection monitors (as indicated by manufacturers).

Equipment	Hydrogen (H${}_{2}$)	Carbon Monoxide (CO)	Moisture	Other Gases
Calisto 2	x	x	x	
SmartDGA Guard	x	x	x	C${}_{2}$H${}_{2}$ and CO${}_{2}$
Minitrans	x	x	x	C${}_{2}$H${}_{2}$
Hydran 201T*i*	x			
Hydran M2-X	x	${}^{1}$	x	
MHT410	x		x	
CoreSense	x		x	
DGA 150	x			
Serveron TM1	x		optional	
Hydrocal 1003	x	x	x	
Hydrocal 1004 GenX	x	x	x	C${}_{2}$H${}_{2}$
Hydrocal 1005	x	x	x	C${}_{2}$H${}_{2}$ and C${}_{2}$H${}_{4}$
H2 Guard	x			
Multisense 5	x	x	x	C${}_{2}$H${}_{2}$ and C${}_{2}$H${}_{4}$

${}^{1}$: Relative sensitivity (15±4%).

**Table 7 sensors-19-04057-t007:** Hydrogen concentration measurement range, accuracy and repeatability of fault detection monitors (as indicated by manufacturers).

Equipment	Measurement Range (ppm)	Accuracy	Repeatability
Calisto 2	2–50.000	±2 ppm or ±5%	±2 ppm or ±3%
SmartDGA Guard	5–10.000	±5 ppm or ±5%	–
Minitrans	5–5.000	±5 ppm or ±10%	–
Hydran 201T*i*	25-2.000	±25 ppm or ±10%	±10 ppm or ±5%
Hydran M2-X	25–2.000	±25 ppm or ±10%	±10 ppm or ±5%
MHT410	0–5.000	±25 ppm or ±20%	±15 ppm or ±10%
CoreSense	0–5.000	±25 ppm or ±20%	–
DGA 150	50–5.000	±25 ppm or ±20%	±15 ppm or ±10%
Serveron TM1	20–10.000	±20 ppm or ±15%	±5 ppm or ±10%
Hydrocal 1003	0–2.000	±25 ppm or ±15%	–
Hydrocal 1004 GenX	0–6.000	±20 ppm or ±10%	–
Hydrocal 1005	0–2.000	±25 ppm or ±15%	–
H2 Guard	25–5.000	±25 ppm or ±20%	±15 ppm or ±10%
Multisense 5	0–2.000	±LDL ppm or ±15%	–

–: Not indicated by the manufacturer.

**Table 8 sensors-19-04057-t008:** Carbon monoxide concentration measurement range and accuracy of fault detection monitors (as indicated by manufacturers).

Equipment	Measurement Range (ppm)	Accuracy	Repeatability
Calisto 2	25–100.000	±25 ppm or ±15%	±25 ppm or ±10%
SmartDGA Guard	10–10.000	±10 ppm or ±5%	–
Minitrans	10–50.000	±10 ppm or ±10%	–
Hydran 201T*i*			
Hydran M2-X			
MHT410			
CoreSense			
DGA 150			
Serveron TM1			
Hydrocal 1003	0–2.000	±25 ppm or ±20%	–
Hydrocal 1004 GenX	0–6.000	±5 ppm or ±10%	–
Hydrocal 1005	0–2.000	±25 ppm or ±20%	–
H2 Guard			
Multisense 5	0–5.000	±LDL ppm or ±5%	–

–: Not indicated by the manufacturer; Blank spaces indicate what the monitors do not measure.

**Table 9 sensors-19-04057-t009:** Moisture measurement range and accuracy of fault detection monitors (as indicated by manufacturers).

Equipment	Measurement Range	Accuracy	Repeatability
Calisto 2	2–100%	±3 ppm or ±3%	±2 ppm or ±2%
SmartDGA Guard	1–99%	±3 ppm or ±2%	–
Minitrans	0–100%	±2%	–
Hydran 201T*i*			
Hydran M2-X	0–100%	±2%	±2%
MHT410	0–100%	±2%	–
CoreSense	0–100%	±2%	–
DGA 150			
Serveron TM1	0–100%	±5%	–
Hydrocal 1003	0–100%	±3 ppm or ±3%	–
Hydrocal 1004 GenX	0–100%	±3 ppm or ±3%	–
Hydrocal 1005	0–100%	±3 ppm or ±3%	–
H2 Guard			
Multisense 5	0–100%	±LDL ppm or ±3%	–

–: Not indicated by the manufacturer; Blank spaces indicate what the monitors do not measure.

**Table 10 sensors-19-04057-t010:** Parameters measured by fault diagnosis monitors (as indicated by manufacturers).

Equipment	5 Gases	7 Gases	9 Gases	Moisture
Calisto 5	x			x
Calisto 9			x	x
SmartDGA Guide			x	x
Taptrans			x	x
Transfix			x	x
Multitrans			x	x
DGA 900			x	x
OPT100		x		x
CoreSense M10${}^{1}$				x
Serveron TM3${}^{2}$				optional
Serveron TM8			x	optional
Hydrocal 1008		x		x
Hydrocal 1009${}^{3}$				x
Multisense 9${}^{3}$				x
TOTUS G5	x			x
TOTUS G9		x		x

5 gases: H${}_{2}$, CO, CH${}_{4}$, C${}_{2}$H${}_{2}$ and C${}_{2}$H${}_{4}$; 7 gases: 5 gases + C${}_{2}$H${}_{6}$ and CO${}_{2}$; 9 gases: 7 gases + O${}_{2}$ and N${}_{2}$; ${}^{1}$: 7 gases + C${}_{3}$H${}_{6}$ and C${}_{3}$H${}_{8}$; ${}^{2}$: measures the gases used in DTM; ${}^{3}$: 8 gases (7 gases + O${}_{2}$).

**Table 11 sensors-19-04057-t011:** Gas concentration and moisture measurements range, accuracy and repeatability of the fault diagnosis monitors (as indicated by the manufacturers).

Equipment	H${}_{2}$ Range (ppm)	CO Range (ppm),	CH${}_{4}$ Range (ppm),	C${}_{2}$H${}_{2}$Range (ppm),	C${}_{2}$H${}_{4}$Range (ppm),	C${}_{2}$H${}_{6}$Range (ppm),	CO${}_{2}$Range (ppm),	O${}_{2}$ Range (ppm)	N${}_{2}$ Range (ppm),	C${}_{3}$H${}_{6}$Range (ppm),	C${}_{3}$H${}_{8}$Range (ppm),	Moisture Range,
	Accuracy and Repeatability	Accuracy and Repeatability	Accuracy and Repeatability	Accuracy and Repeatability	Accuracy and Repeatability	Accuracy and Repeatability	Accuracy and Repeatability	Accuracy and Repeatability	Accuracy and Repeatability	Accuracy and Repeatability	Accuracy and Repeatability	Accuracy and Repeatability
Calisto 5	0–20.000 ±0.5 ppm or ±5% ±0.5 ppm or ±3%	0–30.000 ±10 ppm or ±5% ±10 ppm or ±3%	0–100.000 ±0.2 ppm or ±5% ±0.2 ppm or ±3%	0–100.000 ±0.2 ppm or ±5% ±0.2 ppm or ±3%	0–200.000 ±0.2 ppm or ±5% ±0.2 ppm or ±3%							2–100% ±3 ppm or ±3% ±2 ppm or ±2%
Calisto 9	0–20.000 ±0.5 ppm or ±5% ±0.5 ppm or ±3%	0–30.000 ±10 ppm or ±5% ±10 ppm or ±3%	0–100.000 ±0.2 ppm or ±5% ±0.5 ppm or ±3%	0–100.000 ±0.2 ppm or ±5% ±0.5 ppm or ±3%	0–200.000 ±0.2 ppm or ±5% ±0.5 ppm or ±3%	0–200.000 ±0.2 ppm or ±6% ±0.5 ppm or ±4%	0–100.000 ±15 ppm or ±5% ±15 ppm or ±3%	0–100.000 ±500 ppm or ±15% ±500 ppm or ±10%	0–150.000 ±2.000 ppm or ±15% ±2.000 ppm or ±10%			2–100% ±3 ppm or ±3% ±2 ppm or ±2%
SmartDGA Guide	5–10.000 ±5 ppm or ±5% –	10–10.000 ±10 ppm or ±5% –	2–50.000 ±2 ppm or ±5% –	0.5–10.000 ±0.5 ppm or ±5% –	2–50.000 ±2 ppm or ±5% –	2–20.000 ±2 ppm or ±5% –	10–20.000 ±10 ppm or ±5% –	500–50.000 ±500 ppm or ±5% –	5.000–100.000 ±5.000 ppm or ±5% –			1–99% ±3 ppm or ±2% –
Taptrans	5–5.000 ±5 ppm or ±5% –	2–50.000 ±2 ppm or ±5% –	2–50.000 ±2 ppm or ±5% –	0.5–50.000 ±0.5 ppm or ±5% –	2–50.000 ±2 ppm or ±5% –	2–50.000 ±2 ppm or ±5% –	20–50.000 ±20 ppm or ±5% –	100–50.000 ±10% –	10.000–100.000 ±15% –			0–100% ±3% –
Transfix	5–5.000 ±5 ppm or ±5% –	2–50.000 ±2 ppm or ±5% –	2–50.000 ±2 ppm or ±5% –	0.5–50.000 ±0.5 ppm or ±5% –	2–50.000 ±2 ppm or ±5% –	2–50.000 ±2 ppm or ±5% –	20–50.000 ±20 ppm or ±5% –	100–50.000 ±10% –	10.000–100.000 ±15% –			0–100% ±3% –
Multitrans	5–5.000 ±5 ppm or ±5% –	2–50.000 ±2 ppm or ±5% –	2—50.000 ±2 ppm or ±5% –	0.5-50.000 ±0.5 ppm or ±5% –	2–50.000 ±2 ppm or ±5% –	2–50.000 ±2 ppm or ±5% –	20–50.000 ±20 ppm or ±5% –	100–50.000 ±10% –	10.000–100.000 ±15% –			0–100% ±3% –
DGA 900	5–5.000 ±5 ppm or ±5% < 3%	1–50.000 ±1 ppm or ±3% < 2%	2–50.000 ±2 ppm or ±3% < 2%	0.5–50.000 ±0.5 ppm or ±3% < 2%	1–50.000 ±1 ppm or ±3% < 2%	1–50.000 ±1 ppm or ±3% < 2%	20–50.000 ±20 ppm or ±3% < 3%	100–50.000 ±100 ppm or ±5% < 2%	10.000–100.000 ±15% –			0–100% ±3% <3%
OPT100	0–5.000 ±25 ppm or ±20% ±15 ppm or ±10%	0–10.000 ±10 ppm or ±10% ±10 ppm or ±5%	0–10.000 ±10 ppm or ±5% ±10 ppm or ±5%	0–5.000 ±1 ppm or ±10% ±1 ppm or ±10%	0–10.000 ±10 ppm or ±10% ±10 ppm or ±5%	0–10.000 ±10 ppm or ±10% ±10 ppm or ±5%	0–10.000 ±10 ppm or ±10% ±10 ppm or ±5%					0–100% ±2 ppm or ±10% –
CoreSense M10	25–5.000 ±25 ppm or ±20% –	2–5.000 ±2 ppm or ±5% –	1–10.000 ±1 ppm or ±5% –	0.5–10.000 ±0.5 ppm or ±5% –	2–10.000 ±2 ppm or ±5% –	2–10.000 ±2 ppm or ±6% –	5–20.000 ±5 ppm or ±5% –			10–10.000 ±10 ppm or ±5% –	10–10.000 ±10 ppm or ±5% –	0–100% ±3 ppm or ±2% –
Serveron TM3			5–7.000 ±5 ppm or ±5% < 1%	1–3.000 ±1 ppm or ±5% < 2%	3–5.000 ±3 ppm or ±5% < 1%							0–100% ±3% –
Serveron TM8	3–3.000 ±3 ppm or ±5% < 2%	5–10.000 ±5 ppm or ±5% < 2%	5–7.000 ±5 ppm or ±5% < 1%	1–3.000 ±1 ppm or ±5% < 2%	3–5.000 ±3 ppm or ±5% < 1%	5–5.000 ±5 ppm or ±5% < 1%	5–30.000 ±5 ppm or ±5% < 1%	30–25.000 ±30 ppm or ±5% < 1%	5.000–100.000 ±5.000 ppm or ±5% < 20%			0–100% ±3% –
Hydrocal 1008	0–2.000 ±25 ppm or ±15% –	0–5.000 ±25 ppm or ±20% –	0–2.000 ±25 ppm or ±20% –	0–2.000 ±5 ppm or ±20% –	0–2.000 ±10 ppm or ±20% –	0–2.000 ±10 ppm or ±20% –	0–20.000 ±25 ppm or ±20% –					0–100% ±3 ppm or ±3% –
Hydrocal 1009	0–10.000 ±25 ppm or ±15% –	0–10.000 ±25 ppm or ±20% –	0–5.000 ±25 ppm or ±20% –	0–10.000 ±5 ppm or ±20% –	0–10.000 ±10 ppm or ±20% –	0–10.000 ±15 ppm or ±20% –	0–20.000 ±25 ppm or ±20% –	0–50.000 ±1.000 ppm or ±10% –				0–100% ±3 ppm or ±3% –
Multisense 9	0–10.000 ±LDL or ±5% –	0–10.000 ±LDL or ±5% –	0–5.000 ±LDL or ±5% –	0–10.000 ±LDL or ±5% –	0–10.000 ±LDL or ±5% –	0–10.000 ±LDL or ±5% –	0–20.000 ±LDL or ±5% –	0–50.000 ±LDL or ±10% –				0–100% ±LDL or ±3% –
TOTUS G5	5–5.000 ±5 ppm or ±15% –	10–20.000 ±10 ppm or ±10% –	30–60.000 ±20 ppm or ±10% –	3–5.000 ±3 ppm or ±5% –	5–90.000 ±5 ppm or ±5% –							0–100% – –
TOTUS G9	0–5.000 ±5 ppm or ±5% –	1–50.000 ±1 ppm or ±5% –	1-50.000 ±1 ppm or ±5% –	0.1–50.000 ±0.1 ppm or ±5% –	1–50.000 ±1 ppm or ±5% –	1–50.000 ±1 ppm or ±5% –	3–50.000 ±3 ppm or ±5% –	100–505.000 ±10% –	10.000–150.000 ±15% –			0–100% – –

–: Not indicated by the manufacturer; Blank spaces indicate what the monitors do not measure.

**Table 12 sensors-19-04057-t012:** Analogue inputs and outputs of gas-in-oil monitors (as indicated by the manufacturers)

Equipment		Analogue Input	Analogue Output
Fault detection monitor	Calisto 2	$1(I_{DC})$ (oil temperature sensor)	$3(I_{DC})$
	Smart DGA Guard	x	x
	Minitrans	$1(I_{DC})$ (load sensor)	x
	Hydran 201Ti	$1(I_{DC})$	x
	Hydran M2-X	$1(I_{DC})$	$1(I_{DC})$
	MHT410	x	$3(I_{DC})$
	CoreSense	$3(I_{DC})$	$3(I_{DC})$
	DGA 150	x	$1(I_{DC})$
	Serveron TM1	$2(I_{DC})$	$3(I_{DC})$
	Hydrocal 1003	$4(I_{DC})$ + $4(I_{DC})$ or $4(V_{DC})$	$4(I_{DC})$
	Hydrocal 1004 GenX	optional ${}^{1}$	optional ^1^
	Hydrocal 1005	optional: $4(I_{DC})$ + $6(I_{AC})$ or $6(V_{AC})$	$5(I_{DC})$
	H2 Guard	$2(I_{DC})$	$3(I_{DC})$
	Multisense 5	optional: $10(I_{DC})$	$5(I_{DC})$
Fault diagnosis monitor	Calisto 5	optional: $2(I_{DC})$	optional: $10(I_{DC})$
	Calisto 9	optional: $2(I_{DC})$	optional: $10(I_{DC})$
	Smart DGA Guide	x	x
	Taptrans	optional ^1^	$8(I_{DC})$
	Transfix	optional ^1^	$8(I_{DC})$
	Multitrans	optional ^1^	$8(I_{DC})$
	DGA 900	optional: $15(I_{DC})$	optional ^1^
	OPT100	x	x
	CoreSense M10	$4(I_{DC})$	$8(I_{DC})$
	Serveron TM3	$3(I_{DC})$	x
	Serveron TM8	$3(I_{DC})$	x
	Hydrocal 1008	optional: $4(I_{DC})$ + $6(I_{AC})$ or $6(V_{AC})$	$8(I_{DC})$
	Hydrocal 1009	optional: $4(I_{DC})$ + $6(I_{AC})$ or $6(V_{AC})$	$9(I_{DC})$
	Multisense 9	optional: $10(I_{DC})$	$10(I_{DC})$
	TOTUS G5	$4(I_{DC})$	-
	TOTUS G9	-	-

x: The monitor does not have analogue inputs or outputs; ^1^: Not specified the number of analogue inputs or outputs; –: Not indicated by the manufacturer.

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
