# Peer review of "Dissolved Gas Analysis Equipment for Online Monitoring of Transformer Oil: A Review"

_sensors, 2019, doi:10.3390/s19194057_

Round 1
Reviewer 1 Report
This paper begins with renewable energy. I think it is not a good idea. In fact, technologies of Dissolved Gas Analysis(DGA) had been focused for many years before renewable energy appeared. As shown in this paper, there are many methods such as GC, FTIR, NIR,PAS, and so on, for DGA. I think the authors should discuss the advantages and disadvantages of every methods in detail. Furthermore, the conclusions should come from the latest papers, reports besides the performance specification of products. For the fault diagnosis, there are also many methods. As a review, I think the advantages and disadvantages of every method should also be compared in detail. The raw data should be obtained from the latest papers, reports, and so on.Author Response
Dear Reviewer, Thank you very much for your comments. We have tried to clarify all your comments and follow your suggestions.
R1_1: This paper begins with renewable energy. I think it is not a good idea. In fact, technologies of Dissolved Gas Analysis (DGA) had been focused for many years before renewable energy appeared.
A: As you point out, dissolved gas analysis technologies were developed and used long before renewable energies appeared. However, it is from the integration of renewable energies, when the dynamic management of distribution and transportation networks began to be widely used, which implies that the assets work under different load levels throughout the life cycle. Due to the variability in the levels of operation of the assets, it is essential to have a better knowledge of the health status of the assets, in this case the transformers, in order to operate the system reliably. Our idea when we wrote this introduction was to establish a correlation between the integration of renewable energies and the dynamic management of the network with predictive maintenance and the state of health of the assets. To avoid issues that deviate the reader from the main aim of the review, we decided to eliminate the issue of renewable energy following your suggestion.
R1_2: As shown in this paper, there are many methods such as GC, FTIR, NIR,PAS, and so on, for DGA. I think the authors should discuss the advantages and disadvantages of every methods in detail. Furthermore, the conclusions should come from the latest papers, reports besides the performance specification of products.
A: As you comment, there are many methods for the detection of gases used by DGA equipment for online monitoring. Following your suggestion, we included a new table 5 in which we discuss the advantages and disadvantages of each method. In addition to this new table, several new references have been included (highlighted in red color).
R1_3: For the fault diagnosis, there are also many methods. As a review, I think the advantages and disadvantages of every method should also be compared in detail. The raw data should be obtained from the latest papers, reports, and so on.
A: As you explain, there are many methods for the faults identification in the transformer insulation. We added the new table 3 in which the main methods for the faults identification are described. In addition, a new paragraph describing the relationship between the selection of a fault identification method and the number of gas concentrations that the monitor needs to measure has been included in the conclusions.
In addition, we included several current references that helped us complete the review (highlighted in red color).
Reviewer 2 Report
The paper gives an extensive overview from the existing overview techniques. The paper is well-written, I have two questions what would be nice to be discussed, at least mentioned in this review paper:
Firstly, this paper considers only the case that the transformers are filled with mineral oil based transformer oils. However, this is a current challenge, how the biodegradable, silicon oil or ester type oil based liqiuds. I t would be nice to read something about that what is the difference, if these type of oils are used in the transformers. Because, this is the case if the transformer is installed on the sea for the mentioned wind farms. The ageing and the thermal properties of these materials are quite different from the standard transformer oils:
Mehta, D. M., Kundu, P., Chowdhury, A., Lakhiani, V. K., Jhala, A. S. "A review on critical evaluation of natural ester vis-a-vis mineral oil insulating liquid for use in transformers: Part 1"IEEE Transactions on Dielectrics and Electrical Insulation, 23(2), pp.873–880, 2016. https://doi.org /10.1109/TDEI.2015.005370
Secondly, and mainly for your further research, the power transformers, mainly above 30 - 40 MVA are customly designed machines. Therefore there are lot of differences between two transformers, which is designed for a windfarm or to a pv farm. These differences shown on a case scenario in the following paper:
Orosz, T., SÅ‘rés, P., Raisz, D. and Tamus, Ádám (2015) “Analysis of the Green Power Transition on Optimal Power Transformer Designs”, Periodica Polytechnica Electrical Engineering and Computer Science, 59(3), pp. 125-131. doi: https://doi.org/10.3311/PPee.8583.
There are many optimization strategies and methods are exist to optimize transformers, some of them tries to consider the cost of the maintenance in the very first design stage, however this is another research question, (not the topic of the paper), how or can can use these information (capitalization of the losses, design differences) during the transformer maintenance.
Amoiralis, E. I., Tsili, M. A., Kladas, A. G. "Transformer Design and Optimization: A Literature Survey", IEEE Transactions on Power Delivery, , 24(4), pp. 1999–2024, 2009. https://doi.org/10.1109/TPWRD.2009.2028763
Orosz, T. (2019) “Evolution and Modern Approaches of the Power Transformer Cost Optimization Methods”, Periodica Polytechnica Electrical Engineering and Computer Science, 63(1), pp. 37-50. doi: https://doi.org/10.3311/PPee.13000
Georgilakis, P. S. "Spotlight on Modern Transformer Design", Springer-Verlag, London, UK, 2009. ht t ps://doi.org /10.1007/978-1-84882- 667- 0
Author Response
Dear Reviewer, Thank you very much for your comments. We have tried to clarify all your comments and follow your suggestions.
R2_1: The paper gives an extensive overview from the existing overview techniques. The paper is well-written, I have two questions what would be nice to be discussed, at least mentioned in this review paper:
Firstly, this paper considers only the case that the transformers are filled with mineral oil based transformer oils. However, this is a current challenge, how the biodegradable, silicon oil or ester type oil based liquids. It would be nice to read something about that what is the difference, if these type of oils are used in the transformers. Because, this is the case if the transformer is installed on the sea for the mentioned wind farms. The ageing and the thermal properties of these materials are quite different from the standard transformer oils:
A: As you comment, there are several types of insulation oil. We mentioned in the review the existence of several types of oil to fill the transformer by adding new references (highlighted in red color), and we indicated that in this review we used only the fault identification methods in mineral oil and DGA equipment that measure the gas concentrations in mineral oil.
R2_2: Secondly, and mainly for your further research, the power transformers, mainly above 30 - 40 MVA are customly designed machines. Therefore there are lot of differences between two transformers, which is designed for a windfarm or to a pv farm. These differences shown on a case scenario in the following paper:
A: As you indicate, the custom design of power transformers, mainly above 30 - 40 MVA, influences the health index calculation, so it is necessary to take into account the differences of the transformers in the limits and weights that have to be applied in the equations and algorithms. This information has been included as a new paragraph in the text.
R2_3: There are many optimization strategies and methods are exist to optimize transformers, some of them tries to consider the cost of the maintenance in the very first design stage, however this is another research question, (not the topic of the paper), how or can can use these information (capitalization of the losses, design differences) during the transformer maintenance.
Amoiralis, E. I., Tsili, M. A., Kladas, A. G. "Transformer Design and Optimization: A Literature Survey", IEEE Transactions on Power Delivery, , 24(4), pp. 1999–2024, 2009. https://doi.org/10.1109/TPWRD.2009.2028763
Orosz, T. (2019) “Evolution and Modern Approaches of the Power Transformer Cost Optimization Methods”, Periodica Polytechnica Electrical Engineering and Computer Science, 63(1), pp. 37-50. doi: https://doi.org/10.3311/PPee.13000
Georgilakis, P. S. "Spotlight on Modern Transformer Design", Springer-Verlag, London, UK, 2009. ht t ps://doi.org /10.1007/978-1-84882- 667- 0
A: This is an interesting question and we fully agree it is out of the scope of this paper. In our case we are only considering power transformers that are already installed. As far as we know these transformers didn’t consider that approach during the design stage. We will take your valuable point of view for a future study.
Reviewer 3 Report
The authors propose a review of the prognosis condition in power transformers. The theme falls into the journal scope.
1. Please, include a list of acronyms at the beginning of the text.
2. The authors state that in page 3, lines 58-61: "Thus, the best way to calculate the transformer HI is to always know as many parameters as possible (Table 1); however, although all the parameters are important, they are often partially unknown, difficult to collect or difficult to automatically incorporate into computer programs". Please provide an example or references to demonstrate this statement.
3. In lines 95-96: "The use of the gas concentrations limits the methods of identification of the faults defined in the standards [33,34] to allow for the detection and identification of early failures". This seems to contradict since the authors claim that Dissolved gas-in-oil analysis is one of the methods to be used. Please clarify the point.
4. A description of the theoretical basis of the sensing principle the Dissolved gas-in-oil analysis uses will be a highly desired contribution.
Author Response
Dear Reviewer, Thank you very much for your comments. We have tried to clarify all your comments and follow your suggestions.
The authors propose a review of the prognosis condition in power transformers. The theme falls into the journal scope.
R3_1: Please, include a list of acronyms at the beginning of the text.
A: As you indicate, including the list of acronyms at the beginning of the text would be a good idea; however, the journal template indicates the position of the list of acronyms at the end of the document. We will suggest that change to the editor in chief.
R3_2: The authors state that in page 3, lines 58-61: "Thus, the best way to calculate the transformer HI is to always know as many parameters as possible (Table 1); however, although all the parameters are important, they are often partially unknown, difficult to collect or difficult to automatically incorporate into computer programs". Please provide an example or references to demonstrate this statement.
A: Following your suggestion, we included an example that shows this statement. This example summarizes a typical case in which a valuable information (i.e. thermal image of the transformer case and connections) is difficult to manage as an input parameter.
R3_3: In lines 95-96: "The use of the gas concentrations limits the methods of identification of the faults defined in the standards [33,34] to allow for the detection and identification of early failures". This seems to contradict since the authors claim that Dissolved gas-in-oil analysis is one of the methods to be used. Please clarify the point.
A: As you point out, this phrase makes no sense. There was an error in the revision of the English language in the document; the phrase was corrected with the meaning we wanted to transmit.
R3_4: A description of the theoretical basis of the sensing principle the Dissolved gas-in-oil analysis uses will be a highly desired contribution.
A: As you comment, the description of the theoretical basis of the sensing principle that uses the Dissolved gas-in-oil analysis was included in the review through several references that explain that question in detail. In addition, we include a new table 5 in which we show the advantages and disadvantages of the gas detection technologies used by the DGA equipment presented in this review.
Round 2
Reviewer 1 Report
I think this manuscript can be accepted.